# In Vitro Effect of Photodynamic Therapy with Different Lights and Combined or Uncombined with Chlorhexidine on *Candida* spp.

**DOI:** 10.3390/pharmaceutics13081176

**Published:** 2021-07-30

**Authors:** Vanesa Pérez-Laguna, Yolanda Barrena-López, Yolanda Gilaberte, Antonio Rezusta

**Affiliations:** 1Max Planck Institute for Evolutionary Biology, 24306 Plön, Germany; 2Department of Microbiology, Preventive Medicine and Public Health, University of Zaragoza, 50001 Zaragoza, Spain; yolandabl265@gmail.com (Y.B.-L.); arezusta@unizar.es (A.R.); 3Aragon Health Research Institute (IIS Aragón), 50009 Zaragoza, Spain; ygilaberte@salud.aragon.es; 4Department of Dermatology, Miguel Servet University Hospital, 50009 Zaragoza, Spain; 5Department of Microbiology, Miguel Servet University Hospital, 50009 Zaragoza, Spain

**Keywords:** candidiasis, *C. albicans*, antimicrobial photodynamic therapy, methylene blue

## Abstract

Candidiasis is very common and complicated to treat in some cases due to increased resistance to antifungals. Antimicrobial photodynamic therapy (aPDT) is a promising alternative treatment. It is based on the principle that light of a specific wavelength activates a photosensitizer molecule resulting in the generation of reactive oxygen species that are able to kill pathogens. The aim here is the in vitro photoinactivation of three strains of *Candida* spp., *Candida albicans* ATCC 10231, *Candida parapsilosis* ATCC 22019 and *Candida krusei* ATCC 6258, using aPDT with different sources of irradiation and the photosensitizer methylene blue (MB), alone or in combination with chlorhexidine (CHX). Irradiation was carried out at a fluence of 18 J/cm^2^ with a light-emitting diode (LED) lamp emitting in red (625 nm) or a white metal halide lamp (WMH) that emits at broad-spectrum white light (420–700 nm). After the photodynamic treatment, the antimicrobial effect is evaluated by counting colony forming units (CFU). MB-aPDT produces a 6 log_10_ reduction in the number of CFU/100 μL of *Candida* spp., and the combination with CHX enhances the effect of photoinactivation (effect achieved with lower concentration of MB). Both lamps have similar efficiencies, but the WMH lamp is slightly more efficient. This work opens the doors to a possible clinical application of the combination for resistant or persistent forms of Candida infections.

## 1. Introduction

*Candida* spp. are commensal fungal species commonly colonizing human mucosal and skin surfaces, but they may become pathogenic in some particular scenarios such as treatment with antibiotics, immunocompromised patients, etc., producing in these cases infections that range from superficial to severe skin and mucosal lesions, to even systemic invasion at its worst degree [1]. For example, oral candidiasis is the most common opportunistic infection affecting the human oral cavity. It is caused by an overgrowth of *Candida* spp., being the most prevalent *Candida albicans* [2,3].

Due to the recurrence of *Candida* spp. infections, high systemic antifungal therapy have been widely used, thereby antifungal resistances are increasing. Moreover, patient-dependent, interactions with other medical regimens and organ toxicity can happen [4].

Therefore, it is necessary to develop new treatments such as antimicrobial photodynamic therapy (aPDT). It is based on the use of photosensitizing molecules that are excited with visible light of the appropriate wavelength and reacts with the oxygen, generating reactive species of oxygen to destroy the target pathogen [5,6,7].

Superficial wound infections are potentially suitable for treatment by aPDT because of the ready accessibility of these wounds for both topical delivery of the photosensitizer and light, and because of the exposure to oxygen [6,8,9].

Different aPDT studies have demonstrated that *Candida* spp. can be effectively photoinactivated in vitro and in vivo [5,10,11,12,13].

Future directions of aPDT include the combination with antimicrobials in order to enhance the microbial inactivation and prevent the regrowth when the light from aPDT is turned off and the photoinactivation ends. This original approach has already shown significant potential. It could help to implement the use of aPDT and reduce the amount of antimicrobials used and, thus, the multidrug resistance problem [14,15,16].

Chlorhexidine (CHX) is an antiseptic drug, mainly available in over-the-counter products as routine hand hygiene in healthcare personnel, to clean and prepare the skin before surgery, and before injections in order to help reduce the amount of microorganisms that potentially can cause skin infections [17,18,19,20]. CHX gluconate is also available as a prescription mouthwash to treat gingivitis and as a prescription oral chip to treat periodontal disease [21,22,23] and recently against COVID-19 in dentistry [24].

Here, we investigate the aPDT and the CHX uncombined or in combination against *Candida* spp. As a photosensitizing molecule, we use methylene blue (MB), the main member of the phenothiazine family, well known for its ability to produce singlet oxygen when it is irradiated by red light and react with molecular oxygen [6,8]. As a source of irradiation, we use a light-emitting diode (LED) lamp emitting in red or a white metal halide lamp (WMH) that emits at broad-spectrum white light which is comparable to the emission spectrum of daylight.

The aim is to compare the antimicrobial effect of MB-aPDT when a specific irradiation source or a non-specific broad-spectrum source is used to excite different concentrations of MB. Furthermore, the effects of the combination of aPDT with CHX are evaluated.

## 2. Materials and Methods

The procedure used tried to follow the materials and methods of our previous works and was adapted as follows [25,26,27]:

### 2.1. Chemicals, Media, Strains and Light Sources

-Solvent: Distilled water.-Culture Media: Sabouraud dextrose agar (CM0041 Oxoid^®^, Thermo Scientific, Waltham, MA, USA) and Columbia blood agar BA (Oxoid^®^; Madrid, Spain).-Antiseptic: Chlorhexidine (CHX) (CN162301.0, Miclorbic^®^, Madrid, Spain). Stock CHX solution was diluted in distilled water. CHX was applied at a concentration of 10 μg/mL.-Photosensitizer: Methylene blue (MB), (Sigma-Aldrich^®^; Madrid, Spain). Stock MB solution was diluted in distilled water. All solutions were prepared no more than a week prior to use and handled under light-restricted conditions. The concentration ranges from 640 to 0.03 μg/mL were used.-Strains: *C. albicans, C. parapsilosis* and *C. krusei* were acquired from the American Type Culture Collection (ATCC, Rockville, MD, USA). *C. albicans* ATCC 10231, *C. parapsilosis* ATCC 22019 and *C. krusei* ATCC 6258 were used.-Light sources: Light-emitting diode (LED) lamp, Showtec LED Par 64 Short 18 × RGB 3-in-1 LED, Highlite International, emitting at 625 ± 10 nm (power density 7 mW/cm^2^ at a distance between the LEDs and the microtiter plate with the microbial suspension of 17 cm where the diameter of the light beam is approximately 25 cm) and white metal halide lamp (WMH), made by the Department of Applied Physics of the University of Zaragoza, Spain, emitting at 420–700 nm (power density 90 mW/cm^2^ at a distance between the lamp and the 96-well microtiter plate of 10 cm where the diameter of the light beam is approximately 21 cm). Appendix A shows the lamps and their emission spectrums. Both were used at a fluence of 18 J/cm^2^. This fluence corresponds to a 42.86 min (≈43 min) irradiation time for the samples using the red-LED lamp and 3 min and 25 sec for the samples irradiated with WMH lamp.

### 2.2. In Vitro Photodynamic Treatment of Yeast Suspension

*C. albicans*, *C. parapsilosis* or *C. krusei* seeded on Sabouraud dextrose agar were cultured aerobically overnight at 35 °C. The inoculum was prepared in distilled water and adjusted to 5 ± 0.03 on the McFarland scale (concentrations in the range of >1 × 10^6^ colony-forming units (CFU) per 100 µL and was deposited into 96-well microtiter plates. Two-fold serial dilutions concentrations from 640 μg/mL to 0.03 μg/mL of the MB were added, in absence or presence of 10 μg/mL of CHX (MB+/CHX−/light+) (MB+/CHX+/light+). The final volume in each well was 100 μL. Irradiation proceeded with no preincubation period; the suspensions were immediately subjected to irradiation with fluence of 18 J/cm^2^ using the red-LED lamp or the broad spectrum-WMH lamp. Control samples were subjected to identical treatment, in the absence or presence of the photosensitizer, and were either kept in darkness or irradiated to evaluate the effect of each parameter: negative or initial control (MB−/CHX−/light−), irradiation control (MB−/CHX−/light+), control of photosensitizer in darkness (MB+/CHX−/light−) and antiseptic controls (MB−/CHX+/light−) (MB−/CHX+/light+). After completing the aPDT protocol, samples and controls were assessed in serial dilutions of each suspension and were cultured on blood agar and incubated overnight at 35 °C. The dilutions were made and aliquots were cultured to have blood agar plates with a number of CFUs in the range of 0 to 200 per plate in order to be able to count them reliably.

### 2.3. Efficacy

The efficacy of aPDT treatment was assessed by counting the number of CFU/100 µL using a Flash and Go automatic colony counter (IUL S.A., Barcelona, Spain). A reduction of 6 log_10_ in the number of CFU/100 µL was considered indicative of fungicidal activity. The minimum concentration of MB that reduced yeast survival by 3 log_10_ was also evaluated. All experiments were carried out at least five times. The results are expressed as mean and standard deviation.

## 3. Results

### 3.1. Photoinactivation of Yeasts by MB-aPDT (MB+/CHX−/Light+)

MB-aPDT effectively inactivated *Candida* spp. achieving a reduction of 6 log_10_ in the number of CFU/100 μL in all the studied strains (Figure 1). The minimum concentration of MB required to achieve this effect was 320 μg/mL in all cases except in those irradiated with a WMH lamp in *C. parapsilosis* that required 80 μg/mL and in *C. krusei* between 320–640 μg/mL (Table 1). Analyzing in more detail the sensitivity of each strain to MB-aPDT, *C. krusei* is the most resistant and *C. parapsilosis* and *C. albicans* show a very similar ratio of response, although *C. parapsilosis* is slightly more sensitive to white light than *C. albicans* (Figure 1 and Appendix A).

### 3.2. Fungicidal Effect of MB-aPDT Combined with CHX (MB+/CHX+/Light+)

The antimicrobial effect of MB-aPDT on *Candida* spp. was maintained in the presence of CHX, as evidenced by the 6 log_10_ reduction in the number of CFU/100 μL in all experiments. Moreover, the combination of MB-aPDT using the WMH lamp + CHX achieves this degree of reduction on *C. albicans* decreasing 4-fold the required photosensitizer concentration (the necessary concentration is 1/4) (Figure 1 and Table 1).

To achieve a 3 log_10_ reduction in the number of CFU/100 μL when MB-aPDT is used in combination with CHX, the required concentration of MB needed is at least half compared to the concentration needed using MB-aPDT alone. The greatest reduction of the photosensitizer concentration (8-fold) is achieved against *C. albicans* using the red-LED lamp. On the other hand, the greatest reduction against *C. parapsilosis* occurs with the WMH irradiation (1/4–1/8 of the initial photosensitizer concentration) (Figure 1 and Table 1).

### 3.3. Control of Inoculum and Toxic Effects of MB (MB+/CHX−/Light−), CHX (MB−/CHX+/Light−) and Irradiation (MB−/CHX−/Light+)

No reduction in the number of CFU/100 µL from the initial inoculums (MB−/CHX−/light−) was observed.

Samples with the different MB concentrations evaluated under the same conditions used in irradiation but keeping it in darkness (MB+/CHX−/light−) (dark MB in Figure 1) show significant reductions at the highest concentrations tested as follows: reductions of up to a maximum of 3.5 log_10_ in *C. albicans*, 4 log_10_ in *C. parapsilosis* and 4.5 log_10_ in *C. krusei* were achieved by 640 μg/mL of MB. In all experiments, the effects of keeping the microbial suspension with the different MB concentrations in dark (light−) for 43 min or 3 min and 25 sec (using the time of the irradiation with the red-LED or the WMH lamp respectively) is similar, except for *C. krusei* (reduction of 4.5 log_10_ after 43 min vs. 2.5 log_10_ after 3 min and 25 sec) (Figure 1).

The irradiation with the red-LED lamp or with WMH lamp in the absence of photosensitizer and antimicrobial (MB−/CHX−/light+) did no significantly reduce the number of yeasts (reduction ≤0.3 log_10_, Figure 1).

In the absence of photosensitizer and irradiation, the tested concentration of CHX (10 µg/mL) (MB−/CHX+/light−) (dark MB-CHX with the value of 0 MB concentration in Figure 1) failed to effectively inactivate the yeast. A maximum reduction of 1 log_10_ was observed against *C. parapsilosis.*

The cumulative effect of CHX and irradiation (MB−/CHX+/light+) (Figure 1) reaches a maximum reduction in the number of CFU/100 μL of 1.3 log_10_ against *C. parapsilosis* being the most sensitive strain to this effect.

## 4. Discussion

MB-aPDT is effective in eradicating *Candida* spp. (>6 log_10_ reduction in the number of CFU/100 μL of *C. albicans*, *C. parapsilosis* or *C. krusei*) and the combination with CHX enhances the photoinactivation, i.e., the effect is achieved with lower aPDT-dose (Figure 1 and Table 1).

Regarding the comparison of MB-aPDT results obtained with those reported by other authors, many variables should be considered. Table 2 summarizes different studies against *Candida* spp. in suspension, specifying the methodology and results. Daliri et al. reported a reduction of 3.43 log_10_ of *C. albicans* using 200 μg/mL of MB which is notably lower than the one reached in the present study (MB concentration range of 80–160 μg/mL is able to inhibit 4 log_10_). They used a bigger number of CFU in the inoculum and this could affect but the mismatch may be because they use a laser irradiation source [28]. Application times are usually short when lasers are used and it does not always guarantee adequate oxygenation [6]. Valkov et al. report the absence of effect of MB-aPDT using very low MB concentration (<2 μg/mL) [13]. Ferreira et al. and de Oliveira-Silva et al. achieved different reductions of *C. albicans*, 0.5 log_10_ and 6 log_10_ respectively, with 32 μg/mL of MB and a fluence of 30 J/cm^2^ for 3–4 min. This shows the variability between experiments [29,30]. The results of Ferreira et al. are closer to those of this work (32 μg/mL of MB does not produce complete photoinactivation) (Table 2).

Focusing on *C. parapsilosis*, Güzel Tunçcan et al. achieved a reduction of 4 log_10_ with 25 μg/mL of MB. The comparison with our data and the possible explanation is very difficult because the methodology used is dissimilar [31]. Černáková et al. demonstrated that using 9.6 μg/mL of MB inhibited between 1.13–1.27 log_10_ of *C. parapsilosis* in suspension, similar results to this work (this concentration does not generate complete photoinactivation) [32]. Finally, Ahmed et al. used 100 μg/mL of MB and achieved reductions of 0.58–0.85 log_10_ at fluences of 90–180 J/cm^2^ respectively [33]. Again, the difference may be due to the fact that they used a laser irradiation source and therefore it would be less effective (Table 2).

Against *C. krusei*, concentrations of 16 μg/mL [34] or150 μg/mL of MB [35] even at high fluences only achieves a maximum reduction of 0.65 log_10_. More similar result to ours was obtained by Souza et al. using 100 μg/mL with a reduction of 1.54 log_10_ using a fluence of 28 J/cm^2^ [36]. All MB-aPDT studies together lead us to conclude that *C. krusei* is the most resistant *Candida* spp. to MB-aPDT as well as it is more resistant to antifungals in general mainly due to the characteristics of its membrane [37] (Table 2).

Regarding the MB-aPDT combination with CHX, it stands out that >6 log_10_ reduction in the number of CFU/100 μL of *C. albicans* was achieved reducing the concentration of photosensitizer needed from 320 to 80 μg/mL when WMH lamp was used (Figure 1, Table 1 and Appendix A). Furthermore, the addition of CHX halved the concentration of MB required to reach a reduction of 3 log_10_ in *C. albicans* and *C parapsilosis*, and slightly less than half against *C. krusei*. Therefore, a synergistic effect is seen between MB at concentrations unable to achieve complete photoinactivation and CHX. These results are relevant because the presence of CHX could help to avoid the microbial regrowth of those microorganisms not completely destroyed when PDT is finished. This is one of the disadvantages of using aPDT for infections in the clinic, the risk of microbial regrowth after its application. The combination with antimicrobials could play a crucial role to overcome this limitation of aPDT in this context [14,15].

To our knowledge, there are not studies combining aPDT plus CHX in vivo against *Candida* spp. Recently, the effectiveness of MB-aPDT combined with CHX and zinc oxide ointment has been studied on wound healing process after rumenostomy. This study in cattle ratifies the use of aPDT and suggests that it could be performed for other surgical procedures as a complementary approach or an alternative for topical administration of antibiotics [38]. The combination of CHX plus aPDT has been tried against other microorganisms such as *Porphyromonas gingivalis* biofilm on a titanium surface in a dental framework. The application of CHX and subsequent aPDT using toluidine blue O was shown to be an efficient method to reduce *P. gingivalis* in titanium surfaces [39]. Regarding other studies of antimicrobials plus aPDT against *Candida* spp., Giroldo et al. demonstrated that yeasts, both in suspension and in biofilms, were much more susceptible to antifungal treatments after MB-aPDT, explained by the increase of membrane permeability caused by aPDT [40]. Regarding the in vitro combination of MB-aPDT with fluconazole against resistant strains of *C. albicans, C. glabrata* and *C. krusei*, a synergistic effect was found in fluconazole resistant strains of *C. albicans* and *C. glabrata*, but not against *C. krusei*. [34]. These results do not agree with those found by Snell et al. They showed that fluconazole did not increase the aPDT inactivation of *C. albicans* using MB or another photosensitizer of the protoporphyrin family. However, miconazole did enhance the fungicidal activity of aPDT [41]. Moreover, to our knowledge, there are no studies using aPDT in combination with antimicrobials that report antagonistic effects, which support the use of aPDT in combination with CXH due to the possible advantages [15].

Considering clinical practice, MB-aPDT (660 nm and 7.5 J/cm^2^) has been tried in HIV patients diagnosed with oral candidiasis comparing it with an antifungal commonly used in candidiasis. After 30 days, the antimicrobial was effective, but there were recurrences except when 450 μg/mL of MB was used [42].

All together indicates that aPDT or antimicrobial alone may not be entirely effective against *Candida* spp. that is characterized for causing highly recurrent infection especially in predisposed or immunosuppressed patients. On the other hand, combined treatments such as aPDT plus antimicrobials may prevent recurrent infections and avoid resistance. In addition, the combination in terms of clinical application would decrease the intensity of blue staining caused when the MB is applied on the skin or mucous membranes, making the aPDT procedure more cosmetically appealing.

Regarding the concentration of 10 µg/mL CHX used for this study, it was chosen by taking into account other protocols for the combination of antimicrobials plus aPDT and considering that by itself produces no effect under experimental conditions [25,27], Figure 1 C_0_-CHX. Other studies using other conditions report very different results: e.g., Azizi et al. using 1000 μg/mL of CHX achieved a reduction of 0.71 log_10_ [43]; Do Vale et al. calculated that the minimum inhibitory concentration for *C. albicans* was 3.74 μg/mL using an exposure time of 12–48 h [44]; Ellepola et al. proved that with 50 μg/mL of CHX applied for 30 min achieved 0.38 and 0.5 log_10_ of *C. albicans* and *C. krusei* reduction respectively [45].

Regarding the use of the red-LED lamp or the WHM lamp as a source of irradiation for aPDT, the second proved to be more effective in photoinactivating *Candida* spp. with the exception of against *C. krusei* (Figure 1, Table 1 and Appendix A). The use of LED lamp emitting in red matching the absorption spectra peak of the MB tends to be more efficient in the sense of not wasting irradiation energy and therefore red emission sources are usually the ones chosen for MB-aPDT studies (e.g., shown in Table 2). In addition, red LEDs lamps have added advantages at the time of transferring the use of aPDT to clinical application because they are available in all the PDT clinical units; in addition, these lamps stimulate cellular repair mechanisms in fibroblasts [46] and are already used to treat acne vulgaris, herpes simplex virus infection, shingles, or severe wound healing [6,47].

It is also worth noting the time factor to facilitate the use in the clinic since the WMH lamp needs 3 min and 25 seconds to photoinactivate *Candida* spp. compared to 43 min for the LED lamp, due to the greater irradiance of the former compare to the latter (90 mW/cm^2^ vs. 7 mW/cm^2^ respectively). Furthermore, the experiments were performed without preincubation of the photosensitizer MB with *Candida* spp. prior to irradiation. Andrade et al. and Soria-Lozano et al. demonstrated that a pre-incubation time did not produce greater inactivation of the microorganism, so it is not necessary to add more time to the aPDT procedure [12,48].

On the other hand, a broad-spectrum WMH lamp could be a model of daylight, i.e., which could be used as a source of irradiation for aPDT instead of this lamp. The advantages are that the treatment could be carried out at home and it would require less equipment and personnel (cheaper). However, it also has disadvantages, such as the imprecision in the quantification of the dose of light or duration of exposure, considering that intensity of daylight depends on the season of the year, weather conditions, or geographic location [9,49,50,51]. Another limitation for the use of daylight is the limitation to treat Candida infections not accessible for this light, such as the mouth or the genitalia, which otherwise are the most frequent. Nevertheless, the WMH lamp achieves better results than the LED lamp in this work, demonstrating its efficacy.

Overall, our study aims to open the way for the application of this alternative therapy, MB-aPDT alone or better in combination with CHX, either using lamps with a specific or broad- emission spectrum or even daylight as an irradiation source, to deal with cutaneous and mucosal candidiasis. However, it should be borne in mind that the present findings were obtained following in vitro irradiation of *Candida* spp., therefore clinical studies are required to confirm these results.

## 5. Conclusions

-MB-aPDT is active against *Candida* spp. in water suspension.-CHX enhances the photoinactivation of *Candida* spp. (aPDT plus CHX increases the photoactivity of MB).-White light is a suitable light source for aPDT.-MB-aPDT using a broad-spectrum white light is more efficient than a specific red-LED lamp.-Transfer of this therapy to the clinic could be very convenient.

## Figures and Tables

**Figure 1 pharmaceutics-13-01176-f001:**
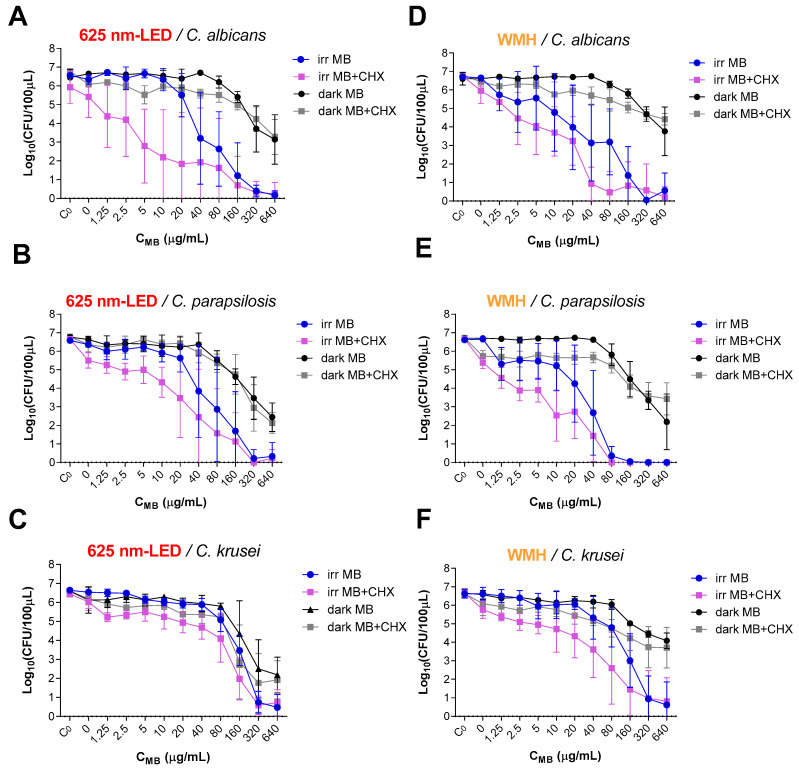
Photoinactivation by antimicrobial photodynamic therapy with methylene blue alone or in combination with chlorhexidine of *C. albicans* (**A**,**D**) *C. parapsilosis* (**B**,**E**) and *C. krusei* (**C**,**F**) using the 625 nm lamp-LED (**A**–**C**) or the WMH lamp (**D**–**F**). The error bars represent the standard deviation calculated for five measurements. C_0_, initial inoculum control; CHX, chlorhexidine; LEDs, light-emitting diodes; MB, methylene blue; WMH, white metal halide.

**Table 1 pharmaceutics-13-01176-t001:** Minimum concentrations of methylene blue (μg/mL) required to reduce the number of *Candida* spp. 3 or 6 log_10_ in the number of colony forming units using the 625 nm-LED lamp or the WMH lamp.

Reduction in the Number of CFU/100 μL	Lamp Used	Treatment	MB Concentration Required for Each Yeast		
			*C. albicans*	*C. parapsilosis*	*C. krusei*
3 log_10_	625 nm LED-lamp	MB-aPDT	40	40–80	160
		MB-aPDT + CHX	5	20	80–160
	WMH lamp	MB-aPDT	40	20–40	80–160
		MB-aPDT + CHX	20	5–10	40–80
6 log_10_	625 nm LED-lamp	MB-aPDT	320	320	320
		MB-aPDT + CHX	320	320	320
	WMH lamp	MB-aPDT	320	80	320–640
		MB-aPDT + CHX	80	80	320–640

aPDT: antimicrobial photodynamic therapy; CHX: chlorhexidine; CFU: colony forming units; LED: light-emitting diodes; MB: methylene blue; WMH: white metal halide.

**Table 2 pharmaceutics-13-01176-t002:** Representative examples reported in literature of planktonic *C. albicans*, *C. parapsilosis* and *C. krusei* photoinactivation caused by MB-aPDT.

Study	Strain	Concentration (μg/mL)	Media	Source and Wavelength (nm)	Fluence (J/cm^2^)	Irradiance (mW/cm^2^)	Initial Load (CFU/mL)	Load Reduction (log_10_)
Güzel Tunçcan et al. (2018) [31]	*C. albicans* ATCC 90028	25	saline	LED-660	0.233	ND	10^6^	3 log_10_
de Oliveira-Silva et al. (2019) [29]	*C. albicans* ATCC 10231	32	PBS	LED-660	10	165	2.5 × 10^6^	0.5 log_10_
					30			6 log_10_
					60			6 log_10_
Ferreira et al. (2016) [30]	*C. albicans* ATCC 90028	32	ND	LED-660	30	250	6.31 × 10^5^	0.5 log_10_
					60			6 log_10_
					120			6 log_10_
Daliri et al. (2019) [28]	*C. albicans* ATCC 10231	100	ND	Laser-660	ND	ND	1.5 × 10^8^	3.3 log_10_
		200						3.43 log_10_
Torres-Hurtado et al. (2019) [52]	*C. albicans*	6.4	PBS	LED-600-650	60	85	2–4 × 10^5^	>5 log_10_
Souza et al. (2010) [53]	*C. albicans* ATCC 18804	100	saline 0.85%	Laser-660	39.5	92	10^6^	3 log_10_
Peloi et al. (2008) [54]	*C. albicans* ATCC 90028	22.5	saline 0.85%	LED-663	6	ND	1–2 × 10^8^	1.31 log_10_
Souza et al. (2006) [36]	*C. albicans* ATCC 18804	100	saline 0.85%	Laser-685	28	92	10^6^	1.25 log_10_
Valkov et al. (2021) [13]	*C. albicans* ATCC 90028	1.6	saline 0.90%	18 W white luminescent lamp-400–700	27	1.9 ± 0.1	1–3 × 10^6^	0
Soria-Lozano et al. (2015) [12]	*C. albicans* ATCC 10231	160	sterile distilled water	WMH-420-700	37	90	1 × 10^6−7^	5 log_10_
This work	*C. albicans* ATCC 10231	320	sterile distilled water	LED-625	18	7	>10^6^	6 log_10_
		320		WMH-420-700		90		6 log_10_
Güzel Tunçcan et al. (2018) [31]	*C. parapsilosis* ATCC 96142	25	saline	LED-660	0.233	ND	3 × 10^6^	4 log_10_
Černáková et al. (2015) [32]	*C. parapsilosis* ATCC 22019	9,6	ND	LED-576-672	15	1.67	ND	1.16 log_10_
	*C. parapsilosis* 16755/2							1.27 log_10_
	*C. parapsilosis* 21922/1							1.13 log_10_
Ahmed et al. (2016) [33]	*C. parapsilosis*	100	ND	Laser-660	90	300	350	0.59 log_10_
					180			0.85 log_10_
This work	*C. parapsilosis* ATCC 22019	320	sterile distilled water	LED-625	18	7	>10^6^	6 log_10_
		80		WMH-420-700		90		6 log_10_
Lyon et al. (2016) [34]	*C. krusei*	16	ND	ND	ND	200	≈5 × 10^5^	0.25 log_10_
Queiroga et al. (2011) [35]	*C. krusei* (ATCC 6258, ATCC 6358, LM08, LM12, LM120)	150	saline 0.85%	Laser-660	60	1000	6 × 10^5^	0.18 log_10_
					120			0.40 log_10_
					180			0.65 log_10_
Souza et al. (2006) [36]	*C. krusei* ATCC 6258	100	saline 0.85%	Laser-685	28	92	10^6^	1.54 log_10_
This work	*C. krusei* ATCC 6258	320	sterile distilled water	LED-625	18	7	>10^6^	6 log_10_
		320–640		WMH-420-700		90		6 log_10_

CFU: colony forming units; MB: methylene blue; ND: no data; PBS: phosphate buffered saline; LED: light-emitting diode lamp; WMH: metal halide lamp.

## Data Availability

Not applicable.

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
