# Peer review of "In Vitro Effect of Photodynamic Therapy with Different Lights and Combined or Uncombined with Chlorhexidine on Candida spp."

_pharmaceutics, 2021, doi:10.3390/pharmaceutics13081176_

Round 1

Reviewer 1 Report

Review of the article: “In vitro effect of photodynamic therapy with different lights and combined or uncombined with chlorhexidine on Candida spp.

Submission ID pharmaceutics-1324877

The manuscript is on very interesting and important subject – development of new strategies of treatment of fungal infections. Below I have presented some comments that authors should take into account preparing final version of the manuscript.

Abstract

Lines 20-21 – were these strains clinical isolates or reference strains? It should be clearly written

Lines26-27 – “the combination with CHX enhances the effect of photoinactivation” but what exactly was the difference? The MB-aPDT produces a 6 log10reduction – what was the result when combination of aPDT and CHX was applied?

Introduction – This part of manuscript is well prepared. I do not have really important critical remarks.

Line 47 – the authors have written: “photosensitizing drug”. I do not think that agents/substances used in aPDT can be classified as “drugs”.

Lines 57-59 – this sentence should be rephrased (or this information should be presented as two sentences)

Line 49, 59, 63 – please check if this form of citations ([5]-[7]) is in agreement with requirements of the Journal  

Line 67 – it should be justified why methylene blue (MB) was selected as a photosensitizing molecule, other well know substances (e.g. toluidine blue O, and rose bengal) also could be applied. In my opinion selection of only one substance is important drawback of the study.

Materials and methods

Lines 117-119 – I would be grateful for some more details about this part of experiment “After completing the aPDT protocol, samples and controls were assessed in serial dilutions of each suspension and were cultured on blood agar and incubated overnight at 35°C.”

Line 122 – please explain why reduction of 6 log10 in the number of CFU/100 μl was considered as indicative of fungicidal activity.

Results

Results are really interesting and well presented – no critical remarks.

Discussion – well prepared

Final decision – minor revision.

Author Response

Dear Reviewer 1,

Thank you very much for your comments.

The modifications that you indicated have been made and we believe that these improve the paper.

Sincerely,

VPL

Reviewer 2 Report

The paper presents a study related to the antimicrobial effect of Methylene  Blue antimicrobial Phototherapy (aPDT) when a specific irradiation source or a non-specific broad-spectrum source is used to excite different concentrations of MB. The effects of the combination of aPDT with chlorhexidine were evaluated.

In the introductory part, the methylene blue used as a photosensitizer was not addressed at all. It should be discussed briefly, even if the chapter on discussions presents studies that address this topic.

Line 67-69. Please provide citation for „As a photosensitizing molecule, we use methylene blue (MB), the main member of the phenothiazine family, well known for its ability to produce singlet oxygen when it 68 is irradiated by red light and react with molecular oxygen”. 

Line 247 . „has been study” it could be replaced by „has been studied”

Lines 65 and 325. need a full stops „.”

The references numbers from the end of article seems odd since they are written twice.

Other remarks. The paper is clearly presented and the chapter related to the discussions presents a material that seems well documented.

Author Response

Dear Reviewer 2,

Thank you very much for your comments.

The modifications that you indicated have been made and we believe that these improve the paper.

Sincerely,

VPL
